# Digital Biomarkers for Neuromuscular Disorders: A Systematic Scoping Review

**DOI:** 10.3390/diagnostics11071275

**Published:** 2021-07-15

**Authors:** Bo-Young Youn, Youme Ko, Seunghwan Moon, Jinhee Lee, Seung-Gyu Ko, Jee-Young Kim

**Affiliations:** 1Department of Global Public Health and Korean Medicine Management, Graduate School, Kyung Hee University, Seoul 02447, Korea; james_youn@khu.ac.kr (B.-Y.Y.); mshssang@gmail.com (S.M.); 2Department of Preventive Medicine, Kyung Hee University, Seoul 02447, Korea; meyougo@khu.ac.kr (Y.K.); epiko@khu.ac.kr (S.-G.K.); 3Department of Korean Medicine, Graduate School, Kyung Hee University, Seoul 02447, Korea; jinhee2770@hotmail.com; 4Department of Neurology, Cheongna Best Rehabilitation Hospital, Incheon 22883, Korea

**Keywords:** digital biomarker, biomarker, neuromuscular disease, NMD, biosensor

## Abstract

Biomarkers play a vital role in clinical care. They enable early diagnosis and treatment by identifying a patient’s condition and disease course and act as an outcome measure that accurately evaluates the efficacy of a new treatment or drug. Due to the rapid development of digital technologies, digital biomarkers are expected to grow tremendously. In the era of change, this scoping review was conducted to see which digital biomarkers are progressing in neuromuscular disorders, a diverse and broad-range disease group among the neurological diseases, to discover available evidence for their feasibility and reliability. Thus, a total of 10 studies were examined: 9 observational studies and 1 animal study. Of the observational studies, studies were conducted with amyotrophic lateral sclerosis (ALS), Duchenne muscular dystrophy (DMD), and spinal muscular atrophy (SMA) patients. Non-peer reviewed poster presentations were not considered, as the articles may lead to erroneous results. The only animal study included in the present review investigated the mice model of ALS for detecting rest disturbances using a non-invasive digital biomarker.

## 1. Introduction

Reliable biomarkers help to detect disease earlier and help to understand disease progression, thereby enabling early intervention. It is important to note that finding reliable biomarkers is arduous in various neurological disorders. Thus, monitoring their efficacy in clinical trials for developing new therapeutic drugs or treatments ought to be up-to-date.

Neuromuscular disorders (NMDs) are one of the major neurological disorders, and they refer to a diverse group, including all disease entities involving motor neurons (e.g., amyotrophic lateral sclerosis), peripheral nerves (e.g., diabetic neuropathies, Charcot-Marie-Tooth disease), muscles (e.g., muscular dystrophies, myositis), and neuromuscular junctions (e.g., myasthenia gravis) [1]. In general, diseases are distinguished from other neurological disorders in terms of their heterogeneous etiologies and variable phenotypes [1,2]. As some diseases have a low prevalence, their pathophysiology has not been completely understood, and other disorders, such as spinobulbar muscular dystrophy, show a slow clinical presentation worsening over several decades. These clinical features of NMDs put much more significant constraints on finding potential biomarkers compared with other disease groups. Hence, it becomes a substantial obstacle to clinical trials for new treatments or drugs.

The biomarkers of NMDs that have been developed to date can be broadly classified as follows: (1) function rating scales or sensory and motor function and/or reflex tests by trained neurologists; (2) specific antibody titers or protein levels, or genomic markers including DNA or RNA determinants; (3) radiological features through MRI or ultrasonography; and (4) neurophysiological values through nerve conduction study/electromyography, magnetic stimulation, and so on [2]. For example, serum creatine kinase (CK) has been routinely measured when diagnosing a disease involving muscle since an increase in CK indicates damage of muscular fibers, and the CK values tend to decrease over time with progressive loss of muscle fibers [1]. However, CK alone has not been considered a reliable or specific biomarker since CK values are not closely associated with disease severity in myopathies.

Since most other biomarkers require a specialized laboratory, including immunoassay, microarray or gene sequencing, or imaging or neurophysiological facilities, there are limitations to equipping every hospital or clinic with such technical facilities. Examination by trained medical staff, such as with a function rating scale, has interrater or intra-rater variability. In addition, scores may not reflect the patient’s actual condition because function is measured episodically at the time the patient visits the hospital [3], especially in the case of diseases worsening rather slowly. Moreover, since numerous NMDs cause muscle weakness or disability, visiting a hospital is exceptionally time-consuming and labor-intensive for patients and caregivers, which is also a big hurdle. The biggest advantage of digital biomarkers could be their ability to compensate for the shortcomings of the traditional biomarkers.

A digital biomarker can objectively and continuously measure and collect biological, physiological, and anatomical data through digital biosensors [3]. The biggest attraction is that it can more accurately reflect the patient’s condition because it enables changes occurring in patients’ daily lives to be measured in real-time; in other words, it is free from temporal and spatial constraints. Moreover, in a situation where a pandemic crisis triggered by infectious diseases such as COVID-19 might be repeated, the development of digital biomarkers will be indispensable in the medical field.

According to a new market intelligence report by BIS research, “Global Digital Biomarkers Market—Analysis and Forecast 2019–2025”, the global digital biomarkers market generated revenue of USD 524.6 million in 2018 and is estimated to grow to over USD 5.64 billion by the end of 2025 [4].

In the neurology field, efforts to develop a digital biomarker seem to focus on neurogenerative disorders such as mild cognitive impairment, Alzheimer’s disease, or Parkinson’s disease [5,6,7,8]. There is still little experience in applying digital biomarkers in actual clinical practice, but experience is expected to increase explosively in the future in conjunction with artificial intelligence and telemedicine. With that said, this systematic scoping review aimed to summarize the available evidence on the current status of research for digital biomarkers in NMDs to suggest research trends and future directions.

## 2. Materials and Methods

We conducted a scoping review with regard to the Extended Preferred Reporting Items for Systematic Reviews and Meta-Analyses for Scoping Reviews (PRISMA-ScR) [9].

### 2.1. Data Sources and Searching

As it is recommended to search at least two bibliographic databases for conducting a review [10], we systematically searched utilizing the following three databases: PubMed, Embase, and Cochrane Library. The NMD search terms were discussed among the authors and finalized by consulting various neurologists; the three authors (BY, YK, SM) ran several sample searches to see if the keywords were relevant to finding enough studies. The complete search keywords are reported in Appendix A. The final search was performed on 24 May 2021, and the search strategies from each database are reported in Appendix B.

### 2.2. Eligibility Criteria

As the study objective was to understand the current status with regard to digital biomarkers for neuromuscular disorders, all types of peer-reviewed publications, such as, original articles, reviews, clinical trials, editorials, and retrospective and prospective studies were included. Additionally, only studies in English were considered. Articles that did not address neuromuscular disorders or digital biomarkers (or biosensors) were excluded. Moreover, non-peer reviewed studies such as poster and oral presentations were not considered, as the results may be fallacious.

### 2.3. Study Selection Process and Data Extraction

After removing the duplicates and non-English articles, three authors (BY, YK, SM) independently reviewed the rest of studies’ titles and abstracts based on the inclusion and exclusion criteria. Then, the authors met and selected the articles to assess the full text. The full text of each study was further reviewed by the aforementioned authors. The references of the selected articles were also screened for more potential studies. When controversy arose before making a final decision for selection, another reviewer (JYK), a neurologist, was involved. Finally, data extraction was completed by the three reviewers (YK, SM, JL) in the following areas: authors, study design, study setting, sensors, biomarkers, main results, and outcomes. Using the scoping review methodology, critical appraisal was not conducted [11,12].

## 3. Results

The initial search retrieved 195 studies, of which 46 were duplicates. After screening titles and abstracts, the full texts of 29 articles were obtained and assessed for eligibility. After reviewing the full text of 29 studies, including the reference lists, 5 more studies were found to be eligible. Of the 34 studies, 24 did not fulfill the inclusion criteria. The reasons for studies being ineligible were as follows: (1) non-English language studies (*n* = 7); (2) not relevant studies (*n* = 9); (3) studies without clear information to analyze (*n* = 1); and (4) non-peer reviewed articles (*n* = 7). A total of 10 studies were examined: 9 observational studies and one animal study (Figure 1).

### 3.1. Overview of Observational Studies

Of the total nine observational studies, four studies were conducted with amyotrophic lateral sclerosis (ALS) patients, three with Duchenne muscular dystrophy (DMD) patients, and the remaining two studies with spinal muscular atrophy (SMA) patients (Table 1).

First, to summarize the studies for ALS patients, one British research group performed a non-controlled, non-drug study to investigate the feasibility of a novel platform for objective data collection of multiple ALS manifestations, including physical activity, heart rate variability (HRV), and speech characteristics through a wearable sensor [13]. The study comprised two phases: a variable-length Pilot Study Phase for refinement of the equipment and data transmission processes and a 48-week Core Study Phase (25 patients enrolled, including the five patients who progressed from Pilot Study Phase to Core Study Phase). During the Core Study Phase, patients visited a clinical site every 12 weeks to perform various assessments and tasks, and the participants wore a sensor in daily life for approximately three consecutive days every month (home monitoring periods). However, the amount and quality of physical activity home monitoring data and HRV data were lower than anticipated. It was found that most of the participating patients safely used the sensor without any inconvenience in their daily life. This study suggested that the monitoring platform could measure physical activity in patients with ALS in their home environment. In addition, this research group compared longitudinally the measures through the home-monitoring sensors with the gold-standard assessments, including ALS Functional Rating Scale-Revised (ALSFRS-R) score and forced vital capacity. As a result, four activity endpoints (average time spent active in the daytime, percentage of time spent active in the daytime, daytime total activity score, 24 h total activity score) showed a moderate correlation with ALSFRS-R total score and a strong correlation with ALSFRS-R gross motor domain score. Additionally, there was a moderate correlation between speech endpoints and ALSFRS-S bulbar domain scores. These study findings highlighted a promising potential of the biotelemetry platform as an efficient clinical evaluation tool of disease progression in ALS patients.

Stegmann et al. emphasized that bulbar deterioration leads to faster decline and shorter survival in ALS patients, assessed patients’ speech features digitally, and evaluated their sensitivity to detect early changes and track progression [15]. The recruited 65 ALS patients provided daily speech samples at home for three months and twice weekly for an additional six months and ALSFRS-R scores on a weekly basis. Their speech was collected remotely via a mobile application, and the articulatory precision (AP) and speaking rate (SR) were assessed through automated speech analysis. This study demonstrated that AP and SR decline was detected earlier than declines on the ALSFRS-R bulbar subscale, and bulbar-onset participants declined faster in AP and SR than nonbulbar-onset participants. Thus, this study showed the possibility of remotely detecting early speech changes and tracking progression in ALS via automated algorithmic assessment of the remotely collected speech.

Stegmann et al. evaluated the repeatability of acoustic and language features of collected longitudinal speech from three separate samples (healthy controls, ALS patients, ALS patients with suspected frontotemporal dementia) [16]. The acoustic and language features were extracted using open source, including openSMILE, Talk2me, and Praat. Overall, the average repeatability scores of speech features were found to be well below acceptable limits for clinical decision-making. This result suggested that researchers should be cautious when developing digital health models with open-source speech features.

Studies on DMD and SMA patients were conducted from the perspective of improving the problem that the currently used outcome measures lack sensitivity and specificity to detect significant improvements within the first 6–12 months of intervention. Heberer et al. collected spatial–temporal data and quantified kinematics and kinetics at the hip, knee, and ankle of 21 DMD boys between 4 and 8 years old using three-dimensional gait analysis over one year [17]. Between the baseline and post-treatment visits, 12 boys began a corticosteroid regimen (mean duration 10.8 +/− 2.4 months) while 9 boys remained steroid-naïve. Significant between-group differences favoring steroid use were found for primary kinetic outcomes (peak hip extensor moments (*p* = 0.007), duration of hip extensor moments (*p* = 0.007), peak hip power generation (*p* = 0.028)), and spatial–temporal parameters (walking speed (*p* = 0.016) and cadence (*p* = 0.021)). This study indicated that hip joint kinetics could identify weakness in DMD boys and are sensitive to corticosteroid intervention.

Le Moing et al. performed a pilot study in seven non-ambulant DMD patients to demonstrate the feasibility and reliability of physical data recorded with a magneto-inertial sensor, ActiMyo^®^ containing a three-axis accelerometer, a three-axis gyroscope, and a three-axis magnetometer [18]. Four variables representative (of upper limb activity were studied: the rotation rate, the ratio of the vertical component in the overall acceleration, the hand elevation rate, and an estimate of the power of the upper limb. This study demonstrated that the ActiMyo^®^ variables were well representative of movements performed during the tasks and correlated well with the scores obtained using other previously validated tests. In particular, the mean of the rotation rate and mean of the elevation rate had the best reliability scores and correlations with task scores, suggesting that they could be good candidates as potential outcome measures in non-ambulatory patients with DMD. The aforementioned study performed the pilot study in a laboratory setting, and then Lilien et al. explored the digital biomarker in home-based monitoring using a wearable magneto-inertial sensor (VMIS) for 23 ambulant DMD patients to evaluate the motricity of neuromuscular patients due to the difficulty of assessing their reduced movement and their abnormal gaits [19]. The authors demonstrated that a precise estimate of foot trajectory in ambulatory DMD is feasible by using their VMIS, and the device’s variables were correlated with the scores obtained using other previously validated tests and are sensitive to change in the DMD patients over six months. This study suggested that their wearable sensor VMIS can record a set of digital biomarkers in the home environment and can be used to evaluate even the most severely impaired patients and can provide objective and reliable data.

Similar studies have been conducted in patients with SMA. Chen et al. developed a specifically designed and user-friendly game based on the Microsoft Kinect sensor, measuring active upper limb movement [20]. They recruited 18 ambulant SMA type 3 patients and 19 age- and gender-matched healthy controls. The elbow angle and arm lifting angle did not show any difference between SMA patients and healthy controls, whereas the hand velocity was found to discriminate between two groups. This study did not demonstrate that this game design is sensitive enough to capture minor differences or early-stage progression in the high-functioning patient group but suggested that the Microsoft Kinetic sensor has the potential of being developed into a complementary output measure for SMA.

A prospective and longitudinal natural history study of patients with Type 2 and 3 SMA has been undergoing over two years [21]. This research group published the baseline data of 81 patients aged 2 to 30 years, of which 19 are non-sitter SMA Type 2, 34 are sitter SMA Type2, 9 non-ambulatory SMA Type 3, and 19 ambulatory SMA Type 3. Most assessments, including the Motor Function Measure, pulmonary function testing, strength, electroneuromyography, and muscle imaging, discriminated between the four groups well. Additionally, the physical activities of patients were measured by three-dimensional sensors—ActiMyo^®^ device, which continuously monitored linear and rotational arm movements and velocity in the home. Five variables representing upper limb activity were analyzed: the wrist angular velocity, the wrist acceleration, the vertical wrist acceleration against gravity, the power, the percentage of activity time. This baseline study showed that the selected variables of patient upper limb activity in real life not only significantly correlated with motor function measure scores, but also significantly correlated with other variables, suggesting that the variables of their sensor can be helpful for evaluating disease progression in the different functional domains and that it also has the potential to assess fatigue and loss of endurance during daily activities. The two-year study results on evaluating the sensitivity of the studied outcomes and biomarkers to disease progression are pending.

### 3.2. Overview of Included Animal Studies

The only animal study included in the present review investigated the SOD1G93A mice model of ALS for detecting sleep and rest disturbances using a non-invasive digital biomarker [22]. Male and female wild-type (WT) and transgenic (SOD1G93A) littermate mice were transferred to a digital ventilated cage (DVC) rack at the age of seven weeks. The DVC, a home cage monitoring system that enables non-intrusive 24/7 animal activity, was used to detect irregular activity patterns that can potentially be associated with sleep and rest disturbances along with the progression of ALS in the SOD1G93A mouse model. Two mice of the same sex and genotype were housed per cage, then assigned to the following experimental groups: (1) males, WT *n* = 18 (9 cages); (2) males, SOD1G93A transgenic *n* = 18 (9 cages); (3) females, WT *n* = 22 (11 cages); (4) females, SOD1G93A transgenic *n* = 20 (10 cages). The mice were monitored from 7 to 24 weeks of age; in addition, body weight decline and neuromuscular function deterioration were measured by grid hanging, and group strength tests were measured. As the ALS progressed over time in SOD1G93A mice, activity patterns started to become irregular. The Regularity Disruption Index (RDI), a novel digital biomarker, was utilized to quantitatively capture the increasing irregularities of activity patterns. The increase of irregularity in daily activity pattern in TG mice could reflect disturbances in sleep and rest behavior since RDI rose during the early symptomatic stage and paralleled grid hanging, and body weight also declined. Therefore, this study suggests that the RDI metric can capture potential sleep and rest disturbances in ALS models. Thus, it could well be used as a digital biomarker to detect disease-related phenotypes.

## 4. Discussion

The present scoping review investigated the available evidence, aiming to identify digital biomarkers in subjects with NMDs; a total of 10 studies were found for this review. Only one article was an animal study, and four out of the nine studies were conducted on ALS patients. Among diseases belonging to NMDs, ALS is one of few diseases for which a functional rating scale (ALSFRS-R) has been recognized as a gold-standard outcome measure that has been widely used in clinical practice and research [23]. One of the four studies for ALS suggested that the marker obtained through the digital sensor had a good correlation with the ALSFRS-R score [14]; another study, which assessed speech features of ALS patients, stated that the biomarkers obtained from the mobile application could detect earlier disease progression than ALSFRS-R bulbar score [15]. Although it was not a human study, an animal study showed that the index obtained through the digital sensor in SOD1gene-transgenic mice was correlated with the decline of body weight and neuromuscular function. These results would be a good precedent for digital biomarker development in the future.

Based on the studies introduced in this review, it seems that research is being conducted to find biomarkers that can be monitored at home using digital sensors that capture physical activity or movement for diseases such as DMD, SMA, and ALS. Digital biomarkers include all human data that can be measured using digital tools that include portable, wearable, implantable, or digestible devices [24]. It is noteworthy that the most representative examples of a digital biomarker include heart rate and physical activity measured using a portable smart band or smart watch [25].

Since many diseases belonging to NMDs mainly present muscle weakness and chronic disease course, establishing a biomarker that can reflect the functional disability of patients in real-time without the patients’ visiting a hospital will be exceptionally valuable. Through this, the burden of patients and caregivers visiting the hospital will be significantly reduced.

In addition, in some NMD diseases, respiratory failure may be accompanied by respiratory muscle or bulbar muscle weakness, which means poor prognosis [26]. Respiratory muscle weakness can occur with acute or subacute onset or as a chronic progressive presentation, typically starting with nocturnal symptoms, leading to difficulty detecting it in the early stage; furthermore, bulbar weakness increases the risk of aspiration so that early intervention, including nasogastric tube insertion, is required. However, at an early stage, it may not be recognized as quickly because it manifests as voice change or slurred speech [26]. Therefore, if these early symptoms that are easy to miss can be detected through digital biomarkers, the medical professionals will more accurately figure the severity or course of the disease by verifying the patient’s daily living and providing effective interventions at the proper time. With that, medical professionals perhaps should consider biomedical signals that directly or indirectly reflect health conditions such as physical activity, skin conductivity, body temperature, electrocardiogram, heart rate, blood sugar, oxygen saturation, electroencephalography, and electromyography as potential digital biomarkers to further detect symptoms [27].

The development of digital biomarkers may advance the era of telemedicine. In particular, chronic disease requires continuous management and education, and conditions that present with gradual respiratory failure require careful follow-up and long-term treatment [26]. The development of digital biomarkers makes home tele-management possible so that medical staff can evaluate the patient’s condition from a distance and can provide counseling and education at the proper time [28]. Additionally, a recent study showed that digital cardiac biomarkers could make a difference in cardiac response to rehabilitation, suggesting that a more patient-tailored treatment becomes possible using digital biomarkers [29].

There are some limitations to this review. First of all, only articles published in English were included. It might have been better to include other language research to better representing the current evidence. Second, there were not enough results to perform the quantitative synthesis needed to obtain a direct result. Lastly, since this review aimed to survey the current status of digital biomarkers applied in NMDs, the eligibility criteria were defined rather broadly. Hence, future research could define detailed eligibility criteria based on our findings.

Nevertheless, this is the first review investigating digital biomarkers in NMDs. As demand for preventive and precision medicine continues to grow, the need for portable and reliable digital biomarkers will also continue to grow in the medical field. That being said, it is vital to collect and analyze digital signals for research purposes for the betterment of treatments in NMDs.

## 5. Conclusions

The results from this review show the potential use of digital biomarkers for various neuromuscular disorders. Research for digital biomarker development in NMDs is at its initial stage. Even though the published studies so far were unable to obtain satisfactory results, the results were promising that the digital biomarkers could be applied to patients in various aspects. In order to be recognized as a digital biomarker, research clarifying whether the prospective biomarker shows a good correlation with the currently established biomarker and its outcome measures should be conducted first. Furthermore, it is necessary to ensure a degree of safety that the digital biomarker does not cause major side effects and that it provides convenience for patients and caregivers without them experiencing a great burden. Future research should consider applying similar techniques on a large scale to verify a digital biomarker’s effectiveness while improving methodological quality.

## Figures and Tables

**Figure 1 diagnostics-11-01275-f001:**
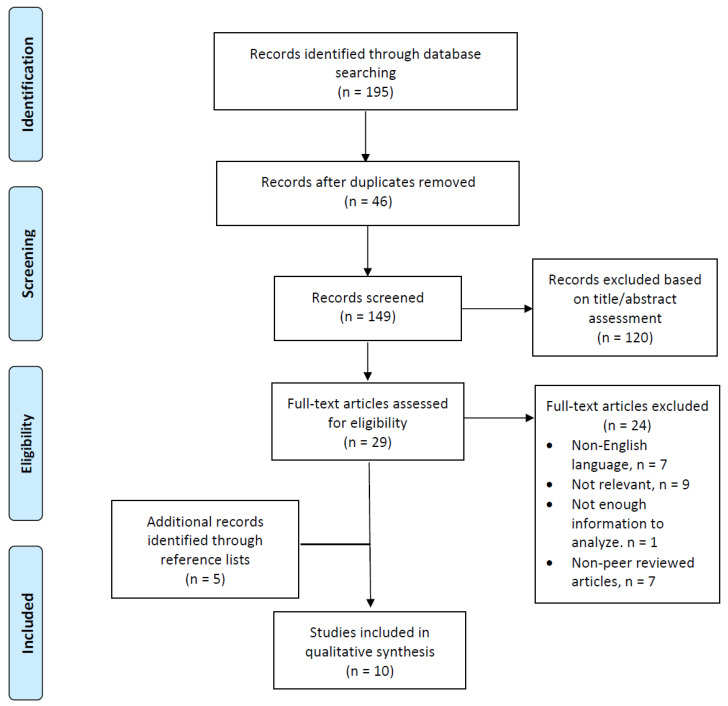
Scoping literature review flow chart (PRISMA).

**Table 1 diagnostics-11-01275-t001:** Summary of findings from retrieved studies.

Authors	Study Design	Study Setting	Sensor	Biomarkers	Main Results	Outcome
Garcia-Gancedo 2019 [13]	Prospective Longitudinal, Cohort Study (variable length pilot study and 48-week core study phase)	ALS patients diagnosed within 18 months of symptom onset (*n* = 25, mean = 53.1 ± 9.93 years)	Home Monitoring Sensor	Physical activity, HRV, digital speech characteristics	A reduction in the patients’ ability to perform activities of daily living over time was observed across all end points. Obtained HRV data were lower than expected. No obvious pattern of speech change over time was observed. There were no serious side effects.	The novel monitoring platform tested in study was successful in collecting ALS patient data, which may be useful in identifying digital markers of disease progression.
Kelly 2020 [14]	Prospective Longitudinal Cohort Study (variable strength pilot study and 48-week core study phase)	ALS patients (*n* = 25, mean age = 53.1 ± 9.93)	Mega Faros 180 accelerometer, 2-lead ECG sensor, bespoke digital speech capture, comparing with ALSFRS-R score	Physical activity (average daytime active, percentage of daytime active, total daytime activity score, total 24 h activity score), HRV, and speech (jitter, shimmer, or speaking rate)	Four physical endpoints showed moderate or strong between patient correlation with ALSFRS-R total and gross motor domain scores.	Four physical activity endpoints showed potential for use as clinical measures of ALS disease progression, using direct, objective, and real-life assessment of physical function.
Stegmann 2020 [15]	Prospective Observational Comparative Study	ALS patients (*n* = 65, mean age = 61 ± 10.2 years) and healthy controls (*n* = 21, mean age = 55 ± 12.5 years)	Mobile application	Articulatory precision (AP), speaking rate (SR)	AP and SR decline was detected earlier than declines on the ALSFRS-R bulbar subscale. AP had significantly decreased as ALS progressed. In the bulbar-onset ALS group, SR showed significant decline.	This study demonstrated that it is possible to remotely detect early speech changes and track speech progression in ALS via automated algorithmic assessment of speech collected digitally.
Stegmann 2020 [16]	Prospective Observational Comparative Study	Sample 1 and 2; ALS patients (*n* = 72, mean age = 59.8 ± 10.4 years) and healthy controls (*n* = 22, mean age = 50.1 ± 14.7 years); sample 3; ALS patients (*n* = 24, mean age = 67.4 ± 11.3 years)	Open-source tool kits (openSMILE, Talk2me, and Praat)	6 acoustic features; energy, frequency, MFCC, pitch, spectral, temporal; 4 language features; lexical, pragmatic, semantic, syntactic	This study evaluated repeatability measures (within-subjects coefficient of variation and intra-class correlation) of acoustic and language features. The repeatability of speech features extracted using open-source tool kits was low.	Researchers should exercise caution when developing digital health models with open-source speech features.
Heberer 2016 [17]	Prospective, Longitudinal Case-Control Study (Baseline and Post-treatment)	DMD patients—steroid group (*n* = 12, mean age = 5.7 ± 1.3) vs. naïve group (*n* = 9, mean age = 5.1 ± 1.1)	Three-dimensional gait analysis	Peak hip extensor moment during stance, duration of the hip extensor moment through stance, peak hip power generation during hip extension	Significant between-group differences favoring the Steroid group were found for peak hip extensor moment, duration of the hip extensor moment, peak hip power generation, and peak ankle power generation.	Hip joint kinetics are early markers of proximal weakness that are responsive to change with corticosteroid intervention, suggesting quantitative gait analysis could play a larger role in the assessment of the efficacy of novel therapeutics.
Le Moing 2016 [18]	Prospective Observational Cohort Study	Non-ambulatory DMD patients (*n* = 7, mean age 18.5 ± 5.5 years	Magneto-Inertial Sensors (ActiMyo^®^)	Angular velocity of the wrist, ratio of the vertical component of the acceleration to the overall acceleration, model-based computed power, elevation rate	The norm of the angular velocity, a model-based computed power, and the elevation rate were significantly correlated with the Minnesota scores and with the writing task.	The mean of the rotation rate and mean of the elevation rate appeared promising since these variables had the best reliability scores and correlations with task scores, suggesting they are good candidates as potential outcome measures in non-ambulant DMD patients
Lilien 2019 [19]	Prospective Observational Cohort Study	DMD patients (*n* = 23, age >5 years)	Wearable Magneto-Inertial Sensor (WMIS)	7 walking parameters and 7 upper limb parameters	The validated 6 min walk test and the North Star Ambulatory Assessment were correlated with their device’s variables and were sensitive to change in the DMD population over a 6-month period.	This study suggests the WMIS can record a set of digital biomarkers and can be used to evaluate even the most severely impaired patients and provides objective and reliable data.
Chen 2017 [20]	Prospective Longitudinal Observational Comparative Study (at baseline, week 12, week 24, week 48)	SMA Type 3 patients (*n* = 18, mean age = 32.3 ± 12.7 years) vs. healthy controls (*n* = 19, mean age = 33.2 ± 13.9 years)	Microsoft Kinect Sensor	Upper limb movement; elbow angle, arm lifting angle, hand velocity	Elbow angle and arm-lifting angle did not show any difference between SMA type 3 patients and controls, hand velocity was faster in SMA patients.	This study suggests that the Microsoft Kinect sensor provides reproducible, objective, and detailed information of body point motion, so has the potential of being developed into a complimentary output measure for SMA.
Chabanon 2018 [21]	Prospective Longitudinal Cohort Study	Type 2 and 3 SMA patients (age 2–30 years); (1) non-sitter SMA Type 2 (*n* = 19), (2) sitter SMA Type (*n* = 34), (3) non-ambulatory SMA Type 3 (*n* = 9), (4) ambulatory SMA Type 3 (*n* = 19) 2 (*n* = 34), (3) non-ambulatory SMA Type 3 (*n* = 9), (4) ambulatory SMA Type 3 (*n* = 19)	Magneto-Inertial Sensors (ActiMyo^®^)	Wrist angular velocity, wrist acceleration, wrist vertical acceleration against gravity, the power, the percentage of activity time	The strongest correlations in this study were observed with the wrist vertical acceleration, and the median wrist angular velocity was decreased in the sitter patients with SMA Type 2 when compared with the non-sitter individuals.	The pending two-year study results will evaluate the sensitivity of the studied outcomes and biomarkers to disease progression.
Golini 2020 [22]	Prospective Observational Comparative Study (Animal Study)	Male and female wild-type (WT) vs. transgenic (SOD1G93A) mice; (1) Males, WT (*n* = 18), (2) Males, TG (*n* = 18), (3) Females, WT (*n* = 22), (4) Females, TG (*n* = 18)	Home cage activity monitoring: Digital ventilated cage (DVC) system; comparing with BW and neuromuscular function	Regularity Disruption Index (RDI)	The rise of RDI in TG mice was remarkable. When computed during daytime. The increase of irregularity in day activity pattern in TG mice could reflect disturbances in their rest/sleep behavior; RDI rose during the early symptomatic stage parallels grid hanging, and BW was declined.	This study suggests that the RDI metric is able to capture potential rest/sleep disturbances in ALS models. Thus, it could be used as a digital biomarker to detect disease-related phenotypes.

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
