# Peer review of "Digital Biomarkers for Neuromuscular Disorders: A Systematic Scoping Review"

_diagnostics, 2021, doi:10.3390/diagnostics11071275_

Round 1

Reviewer 1 Report

In this review article, the authors took an interesting analysis approach to understand the digital biomarkers in neuromuscular disorders based on some peer-reviewed and some non-peer reviewed studies. While the findings are relevant, I am concerned that non-peer reviewed material may lead to erroneous results that will impact the data analysis. I recommend the analysis focussed on the peer-reviewed and or verified information to draw meaningful conclusions.  

Author Response

Dear Reviewer:

First of all, I appreciate your time and help to review my manuscript. 

As you mentioned, we have excluded all non-peer reviewed articles from the manuscript. Therefore, we have restructured the table, deleted the poster presentation section from the Results and revised the reference list. We also revised our abstract and Methods section. Please review the attached file; we have highlighted the changes in yellow.

We again appreciate your time and help in advance,

Bo Youn

Reviewer 2 Report

This paper is well organized and showed a new perspective of biomarkers of neuromuscular junction disorders. The references were proper, and they were reviewed thoroughly.

I concern only about some minor things.

1) The order of references: It will be better if the number of references was numbered according to the order in the text. And some research has not marked a bibliography in the text. Please mark a bibliography of all research in the text.

2) The format of the table

The table is hard to understand. The more organized form will be better. For example, sort according to diseases, and add rows dealing with publication type, the objective of each research- disease progression, comparison of biomarker with other tests, and comparing parameters. The below table is an example for you.

Authors

Study design

Publication type

Study setting

Sensor

Biomarker

Objectives

Comparing parameters

Main results

Outcome

ALS

Original article

Poster

SMA

DMD

IIM

DM neuropathy

3) please find attached the file for other minor concerns.

Author Response

Dear reviewer:

We appreciate your time and help to review our manuscript.

First of all, as reviewer 1 suggested, we have decided to exclude all non-peer reviewed presentations since they may lead to erroneous results so that the we have focused on the 10 peer-reviewed publications for this study.

1) We have re-ordered our reference list for lucid flow of the manuscript for readers.

2) We have restructured our table; however, we have decided to exclude objectives and comparing parameters since the study setting and design explains all. More importantly, the table could not fit within the designated journal's manuscript design. 

3) Please review the attached for minor concerns. We have highlighted the changed made in yellow for easier track and follow.

We again appreciate your time and help in advance. 

Round 2

Reviewer 1 Report

The authors have revised the manuscript analysis based on published data and removing non-peer reviewed material.